# In vitro DNA SCRaMbLE

Yi Wu [1,2,3], Rui-Ying Zhu[1,2], Leslie A. Mitchell[3], Lu Ma[1,2], Rui Liu[1,2], Meng Zhao[1,2,3], Bin Jia [1,2], Hui Xu[1,2], Yun-Xiang Li[1,2], Zu-Ming Yang[1,2], Yuan Ma[1,2], Xia Li [1,2], Hong Liu[1,2], Duo Liu[1,2], Wen-Hai Xiao[1,2], Xiao Zhou[1,2], Bing-Zhi Li[1,2], Ying-Jin Yuan [1,2] & Jef D. Boeke[3]

The power of synthetic biology has enabled the expression of heterologous pathways in cells, as well as genome-scale synthesis projects. The complexity of biological networks makes rational de novo design a grand challenge. Introducing features that confer genetic flexibility is a powerful strategy for downstream engineering. Here we develop an in vitro method of DNA library construction based on structural variation to accomplish this goal. The "in vitro SCRaMbLE system" uses Cre recombinase mixed in a test tube with purified DNA encoding multiple loxPsym sites. Using a β-carotene pathway designed for expression in yeast as an example, we demonstrate top-down and bottom-up in vitro SCRaMbLE, enabling optimization of biosynthetic pathway flux via the rearrangement of relevant transcription units. We show that our system provides a straightforward way to correlate phenotype and genotype and is potentially amenable to biochemical optimization in ways that the in vivo system cannot achieve.

[1] Key Laboratory of Systems Bioengineering (Ministry of Education), School of Chemical Engineering and Technology, Tianjin University, 300072 Tianjin, China. [2] SynBio Research Platform, Collaborative Innovation Center of Chemical Science and Engineering (Tianjin), Tianjin University, 300072 Tianjin, China. [3] Institute for Systems Genetics and Department of Biochemistry and Molecular Pharmacology, NYU Langone Health, New York, NY 10016, USA. These authors contributed equally: Yi Wu, Rui-Ying Zhu. Correspondence and requests for materials should be addressed to J.D.B. (email: Jef.Boeke@nyumc.org)

With the rapid development of DNA synthesis and assembly technologies, there is an emerging use of synthetic DNA for de novo design and construction of heterologous pathways and synthetic genomes[1,2]. However, with increasing biological complexity and the number of genes in the designed system, a major challenge lies in the "debugging" process to ensure that synthetic DNA carries out the intended "designer" function(s)[3–5].

Cre/loxP is a widely used site-specific DNA recombination system derived from bacteriophage P1. Cre recombinase catalyzes a site-specific recombination reaction between two loxP sites and does not require accessory factors[6]. The loxP site is 34 bp in length, consisting of two 13 bp inverted repeats separated by an 8 bp asymmetric spacer sequence. The Cre/loxP system can be used to generate deletions, inversions, insertions (transpositions), or translocations depending on the orientation and location of loxP sites specified in a given system[7]. The simplicity of the Cre/loxP system has led to its use in both in vivo and in vitro applications. Previous in vivo applications include targeted gene knock-out, gene replacement and more[8,9], and in vitro applications comprise high-throughput DNA cloning and adenoviral vector construction[10,11]. The general goal of most existing Cre/loxP applications is to recover a single recombination event at defined positions.

If loxP sites encode a symmetric spacer region (loxPsym), rearrangements are orientation-independent and DNA fragments between two loxPsym sites should undergo deletions or inversions with equal frequency[12,13]. The in vivo Synthetic Chromosome Rearrangement and Modification by LoxPsym-mediated Evolution (SCRaMbLE) system, built into synthetic yeast chromosomes, has been demonstrated to generate stochastic diversity in chromosome structure, including deletions, duplications, inversions, insertions (transpositions), or translocations in synthetic chromosomes synIII and synIXR[13–17]. In this system, the Cre recombinase is introduced into Sc2.0 cells genetically and controlled both transcriptionally and chemically[14,15].

Here, we report an in vitro SCRaMbLE system, driven by recombinant Cre recombinase mixed together in a test tube with purified DNA encoding loxPsym sites. We demonstrate two strategies using the in vitro SCRaMbLE system for pathway engineering and optimization. The top-down method specifies use of a single DNA construct encoding multiple loxPsym sites and the generation of a library of SCRaMbLEd DNA. The bottom-up system consists of an "acceptor vector" with a pool of donor fragments flanked by loxPsym sites. With the addition of Cre recombinase to the reaction, donor fragments are randomly inserted into the acceptor vector to produce a pool of diverse constructs which add one or more donor constructs to the base pathway. The products of both in vitro SCRaMbLE strategies can be transferred to a host strain directly for phenotype testing and genotyping of individual SCRaMbLE derivatives. Using the β-carotene pathway in yeast as an example, we demonstrate how these two in vitro SCRaMbLE strategies can be used for library construction and pathway optimization. Our results indicate that in vitro SCRaMbLE is a unique and straightforward method for generating DNA libraries, and is potentially amenable to biochemical optimization in ways not achievable in vivo.

## Results

**Top-down in vitro SCRaMbLE.** The "top-down" in vitro SCRaMbLE system specifies use of purified Cre recombinase for rearrangement-based optimization of DNA constructs encoding multiple loxPsym sites. The loxPsym sites flank "transcription unit" (TU) sequences, the unit to be SCRaMbLEd in the system. In the presence of Cre recombinase, TUs will be randomly deleted, inverted or duplicated mediated by Cre/loxPsym

reactions. Following transformation of the population of SCRaMbLEd molecules into cells, resultant phenotypes and genotypes can be evaluated and linked (Fig. 1a).

To test the "chemical" feasibility of top-down in vitro SCRaMbLE, 10 loxPsym sites were evenly distributed across a 5 kb piece of DNA and assembled into a plasmid (pYW0261) by overlap polymerase chain reaction (PCR) (Fig. 1b). After a one hour incubation with Cre recombinase, the DNA library was transformed into Escherichia coli to more easily visualize products. To test the diversity of recovered sequences, a pool of SCRaMbLEd plasmids was extracted and then linearized for gel electrophoresis. Nine individual bands were observed, corresponding to the expected sizes for deletions between variously spaced loxPsym sites (Fig. 1c). This is consistent with no obvious preference of recombination between loxPsym sites in the system of in vitro SCRaMbLE. We observed similar results using constructs with 10 loxPsym sites spaced 100 and 1000 bp apart (Supplementary Fig. 1).

To biologically test the system, we performed an in vitro SCRaMbLE experiment with a yeast/E. coli centromeric shuttle vector (pLM495) encoding four β-carotene pathway TUs flanked by five loxPsym sites (Fig. 1d). In this pathway, three carotenogenic genes were sourced from the carotenoid-producing fungus Xanthophyllomyces dendrorhous (crtE, crtI, crtYB)[18] and a truncated HMG1 gene was derived from Saccharomyces cerevisiae. Different promoters and terminators were selected from the S. cerevisiae genome to drive expression of each pathway gene[19]. In vitro SCRaMbLE with Cre recombinase was performed on the purified plasmid for one hour and the products were transformed to E. coli for genotypic testing. DNA purified from E. coli was subjected to digestion and pulsed-field gel analysis, which revealed diverse deletion events for SCRaMbLEd pLM495 (Supplementary Fig. 2). Subsets of plasmids bearing deletions of varied length were isolated and evaluated by gel electrophoresis. To further determine the efficiency of deletions and inversions, we performed PCR analysis of individual E. coli colonies. A total of 300 colonies were randomly picked and analyzed by PCR within individual genes to evaluate deletion frequency. Approximately 27% of the colonies carried at least one deletion event (Supplementary Fig. 3). Another 100 colonies were picked and analyzed by PCR using primers spanning loxPsym sites to evaluate inversion frequency; colonies showing existence of individual genes by PCR but absence of junction regions are inferred to have undergone inversion events. Approximately 28% of the colonies had evidence of inversion events (Supplementary Fig. 4). This result of roughly equal efficiency of deletion and inversion for in vitro SCRaMbLE system is consistent with a previous report for Cre/loxPsym in vivo[12,13]. To test whether the efficiency of in vitro SCRaMbLE was related to the number of loxPsym sites in the substrate DNA, we counted and compared deletion frequencies of in vitro SCRaMbLEd pLM495 or synIXR-BAC, a previously de novo synthesized ~100 kb BAC which encodes 43 loxPsym sites (Supplementary Fig. 5a)[14]. Using PCRTag analysis[14] we observed deletions of DNA segments after in vitro SCRaMbLE (Supplementary Fig. 5b). A total of 46 synIXR-BAC colonies were randomly picked for this analysis, which was carried out by real time PCR[20] (Supplementary Fig. 5c). The deletion frequency for the synIXR-BAC was ~70%, which is higher than with the five loxPsym site plasmid pLM495 (Supplementary Fig. 5d). This suggests that the number of recombination events is positively correlated with loxPsym site number.

We used a single molecule real time sequencing method (Pacific BioSystems SMRT; PacBio) to analyze the diversity of the SCRaMbLEd library recovered from E. coli. PacBio enables PCR-free long read sequencing, which is appropriate to identify

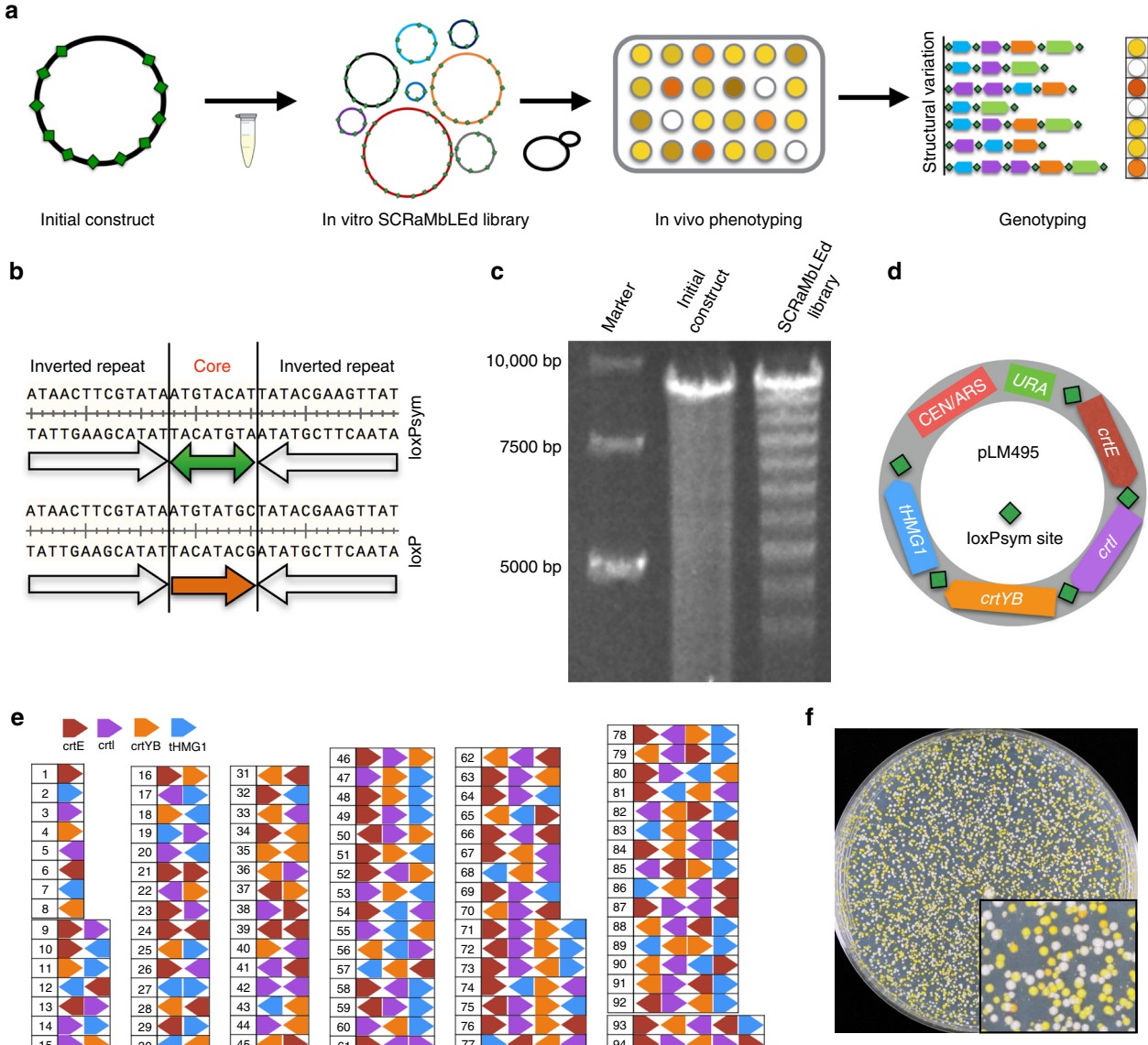

**Fig. 1** Top-down in vitro SCRaMbLE. **a** Schematic of top-down in vitro SCRaMbLE. Green diamonds represent the 34 bp loxPsym site. **b** Sequence comparison between loxPsym and loxP sites. **c** Gel electrophoresis analysis of an in vitro SCRaMbLEd library. The parental construct encoded 10 loxPsym sites with an inter-site distance of 500 bp. Material for linearization with *Not*I was extracted from a pool of *E. coli* colonies carrying the SCRaMbLEd DNA. **d** Map of pLM495. LoxPsym sites flank the β-carotene pathway genes *crtE, crtI, crtYB,* and *tHMG1*. Transcription units for these genes are *pTIP1-crtE-tACS2, pPGK1-crtI-tASC1, pTDH3-crtYB-tCIT1, pZEO1-tHMG1-tACS2*. **e** A total of 94 unique pathway structures were determined by PacBio sequencing of a SCRaMbLEd pLM495 library. **f** Yeast colonies transformed with in vitro SCRaMbLEd pLM495. The magnified region shows different colony colors, consistent with production of colored carotenoid intermediates. Synthetic complete medium lacking uracil (SC–Ura) medium was used to select transformants

structural variation in the DNA library. With only four genes in pLM495, a total of 94 unique constructs were detected in the SCRaMbLEd pool (Fig. 1e). Recombination between multiple loxPsym sites resulted in deletions, inversions, duplications, and other complex combinational events. Considering the limited read depth and low probability of longer DNA reads, we believe that the diversity of SCRaMbLEd molecules is even higher than observed in this experiment.

The SCRaMbLEd product of pLM495 was also directly transformed into *S. cerevisiae* for phenotypic testing. β-carotene production in yeast cells yields yellow colonies, and other pathway intermediates such as lycopene produce other colors[18]. After in vitro SCRaMbLE of pLM495, we saw various colony colors on the yeast transformation plate, including white, yellow, and deep yellow (Fig. 1f). A total of 100 yeast colonies with varied colors were picked randomly and the plasmids were recovered into *E. coli* for PCR analysis and DNA sequencing. Here we identified 17 unique β-carotene pathway structures that included deletion, inversion, and duplication events (Fig. 2 and Supplementary Fig. 6). Yeast cells carrying the 17 unique constructs were tested for β-carotene production using high-performance liquid chromatography (HPLC) (Fig. 2 and Supplementary Fig. 7a). The white yeast strains yYW0408, yYW0213, yYW0410, yYW0411, and yYW0409 lost production of β-carotene due to carotenoid gene deletions. The strains yYW0212, yYW0400, yYW0398, and yYW0399 increased the production of β-carotene with a 3.5–5.1 fold-change, likely the consequence of duplication of *crtI* gene. This result is consistent

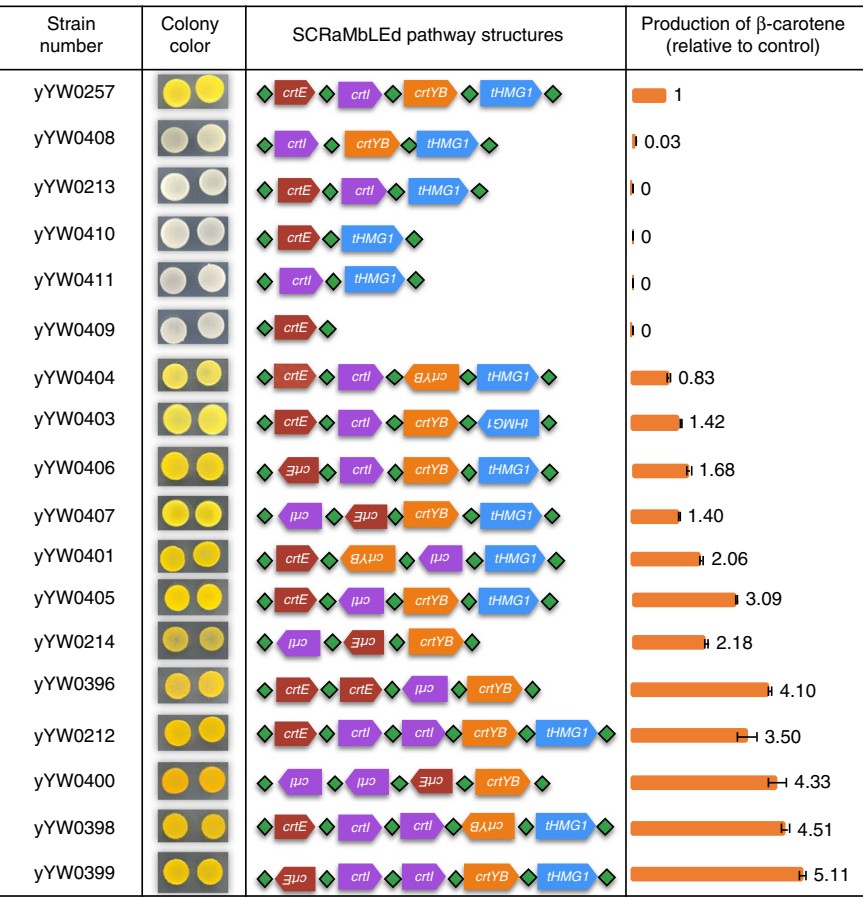

**Fig. 2** Genotype–phenotype analysis of top-down in vitro SCRaMbLEd strains. The colony pictures were taken after 3 days incubation on SC–Ura medium. Strains derive from the experiment in Fig. 1f. The pathway structures of 17 SCRaMbLEd strains were verified by PCR analysis (Supplementary Fig. 6) and Sanger sequencing of the recovered yeast plasmids. The production of β-carotene was determined by high-performance liquid chromatography (HPLC) (Supplementary Fig. 7). Error bars represent standard deviation from three replicates

with a previous report showing an additional copy of *crtI* in yeast leads to the production of higher levels of β-carotene[18]. Interestingly, we found that inversion of *crtI* in strains yYW0401, yYW0405, and yYW0396 also correlated with higher β-carotene production. We evaluated mRNA and DNA level by qPCR analysis of individual genes within the pathway in these strains (Supplementary Fig. 8). The results showed increased *crtI* mRNA levels in the strains yYW0405 and yYW0396 and no obvious changes in the DNA level. This supports the conclusion that *crtI*, encoding a phytoene desaturase, catalyzes the rate-limiting step of this heterologous β-carotene pathway in *S. cerevisiae*. Of all the tested strains, yYW0399 yielded 1.7 μg per mg (dry weight) production of β-carotene, corresponding to a 5.1-fold increase in yield compared to the original construct.

Distinct from traditional mutagenesis, which largely targets the base pair level, in vitro SCRaMbLE provides a simple strategy to mutagenize DNA at the level of structural variation. We compared in vitro SCRaMbLE to two conventional methods for generating libraries, random mutagenesis with error-prone PCR, and atmospheric and room temperature plasma (ARTP)[21,22]. A randomized mutation library of the *crtI* gene in pLM495 was generated with a mutation rate of ~5–10 bp per kb. The randomized library was transformed into *S. cerevisiae* for phenotypic testing. A total of 16 colonies with varied color were screened from 1611 colonies on the plate and then subjected to β-carotene measurements (Supplementary Fig. 9a). Two strains (yYW0429 and yYW0439) showed increased production of β-carotene with 3.1 and 2.4 fold-changes. Of course, these

colonies are also predicted to contain ~10,000 new SNPs, any of which might be deleterious to the production of β-carotene in unanticipated ways. For ARTP, a total of 17 colonies with varied color were screened from 2353 colonies after exposing yeast strain yYW0257 to ARTP jet for 10 and 20 s. Among these, yYW0420 showed a 3.9 fold-change compared with the initial strain (Supplementary Fig. 9b). For the random mutagenesis method, there were many white colored colonies generated indicating a high rate of negative mutation. For ARTP, a lot of treated cells were dead and most of the residual colonies showed unchanged color, indicating a low mutation rate. These results indicated better performance of in vitro SCRaMbLE over two other methods to improve β-carotene production.

**Bottom-up in vitro SCRaMbLE.** "Bottom-up" in vitro SCRaM-bLE starts with a centromeric acceptor vector and a series of "donor fragments"; the basic goal here is to evaluate a series of candidate genes (represented as "donor fragments") for their ability to boost production of the core pathway (resident in the chromosome in a non-SCRaMbLEable format) (Fig. 3a). The donor fragments can consist of the main pathway genes themselves, other genes from the host that produce starting metabolites, or any candidate gene that may positively impact on pathway flux. There are two ways to use the bottom-up system, based on how the selectable markers are exploited. In the first version, (left panel, Fig. 3a), the acceptor vector has two loxPsym sites. The donor fragments are generated from a universal vector,

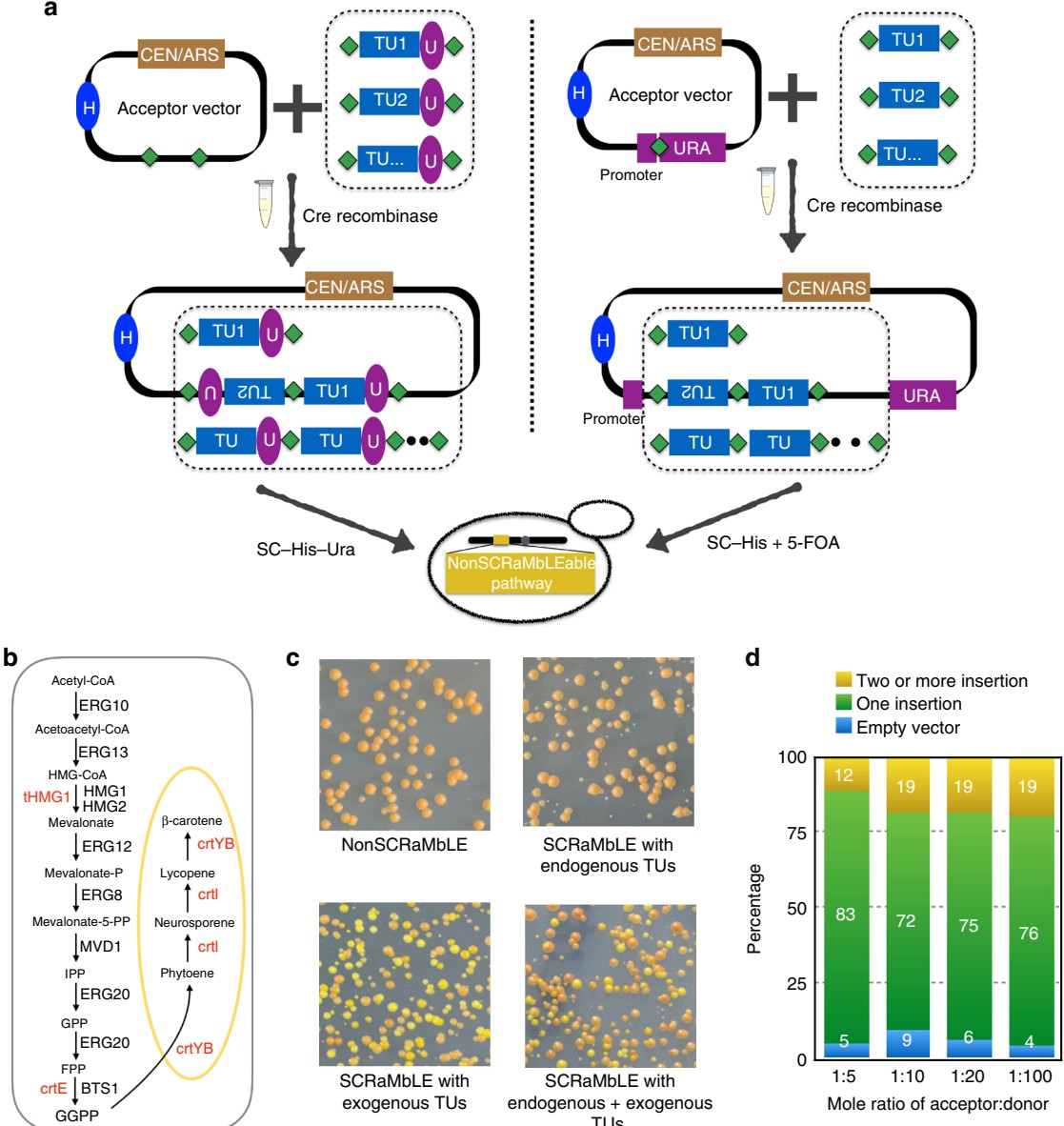

**Fig. 3** Bottom-up in vitro SCRaMbLE. **a** Schematic of two independent bottom-up in vitro SCRaMbLE strategies. Left panel—donor fragments each carry a TU and a URA3 gene. The acceptor vector encodes two loxPsym sites (green diamond). Right panel—a loxPsym site is inserted in frame with the *URA3* coding sequence (Supplementary Fig. 10). Donor fragments are flanked by loxPsym sites. In both cases, the acceptor vector and the pool of donor TUs constructs are mixed with Cre recombinase in vitro. The donor TUs will be randomly inserted into loxPsym sites of the acceptor vector. Note that in bottom-up SCRaMbLE the core β-carotene pathway itself is not in a SCRaMbLE format (i.e., unlike "top down", there are no lox sites flanking core pathway genes). (yellow box "NonSCRaMbLEable pathway" integrated into genome). Transcription unit (TU), *HIS3* auxotroph marker (H), *URA3* auxotroph marker (U). **b** Overview of the carotenoid biosynthetic pathway in *S. cerevisiae*. Genes shown in black are endogenous to *S. cerevisiae*. Genes shown in red are non-native, and derive from *X. dendrorhous* (*crtE*, *crtYB*, *crtI*), and one from *S. cerevisiae* (truncated 3-hydroxy-3-methylglutaryl-coenzyme A reductase gene [*tHMG1*]). **c** Yeast colonies transformed with bottom-up in vitro SCRaMbLEd candidate carotenogenic TU pools. The non-SCRaMbLE sample was transformed with the acceptor vector as a control. Three other in vitro SCRaMbLEd pools consisted of endogenous TUs, exogenous TUs, and endogenous + exogenous TUs as indicated. **d** The efficiency of bottom-up in vitro SCRaMbLE (strategy 1). Different mole ratios of acceptor vector with donor TUs (1:5, 1:10, 1:20, 1:100) were used to test SCRaMbLE efficiency. pYW0113 was used as the acceptor vector; *crtI* TU and *tHMG1* TU were used as donor fragments. A total of 100 yeast colonies for each group were tested using long fragment PCR and restriction enzyme digestion of recovered plasmids

which is an *E. coli*-based plasmid enabled for yeast Golden Gate assembly and red/white *E. coli* colony screening[23]. The donor fragments each encode a *URA3* gene as a positively selectable marker; yeast transformants that are His⁺ Ura⁺ are guaranteed to have picked up at least one donor fragment during the in vitro recombination reaction. In the second version, (right panel, Fig. 3a), the acceptor vector encodes a single loxPsym site, inserted in the *URA3* coding sequence by adding two base pairs

(TG) to the 3′ end of the 34 bp loxPsym site, resulting in an in-frame insertion of 36 bp (Supplementary Fig. 10). A functional Ura3 protein is produced, enabling selection on medium lacking uracil for the parental vector. Recombination of one or more donor fragments into this site physically separates the *URA3* promoter and ATG codon from the coding sequence, enabling negative selection on 5-fluoroorotic acid (5-FOA) medium[24]. Here the donor fragments can be directly amplified by PCR with

primers encoding terminal loxPsym sites. The bottom-up in vitro SCRaMbLE reaction consists of a pool of donor fragments, the acceptor vector, and Cre recombinase. The donor fragments can be heterologous or endogenous transcription units. When the in vitro SCRaMbLEd pool of DNA molecules is transformed to an appropriately engineered host strain with a resident "unSCRaMbLEable" pathway, the addition of one or more candidate TUs will add new genes, and those that augment pathway production can be selected by looking for enhanced color.

Using the β-carotene pathway as an example, we first converted the pathway genes (crtI, crtE, and crtYB) to the unSCRaMbLEable format (no lox sites) and integrated them into

the CAN1 locus (Fig. 3b). We generated seven candidate donor TUs fragments from the mevalonate pathway (ERG10, ERG13, ERG12, ERG8, MVD1, ERG20, and BTS1) and four candidate donor TUs fragments from the exogenous pathway (crtI, crtE, crtYB, and tHMG1) as candidates for bottom-up SCRaMbLE. Using the strategy in the left panel of Fig. 3a, three SCRaMbLEd TU pools (endogenous TUs, exogenous TUs, and all TUs) were transformed into a yeast strain yYW0301 encoding the resident, unSCRaMbLEable β-carotene pathway. After incubation for 3 days at 30 °C, brighter yellow to orange colonies grew only on selective plates carrying SCRaMbLEd exogenous TU pools. There were no distinct color variants on plates with the

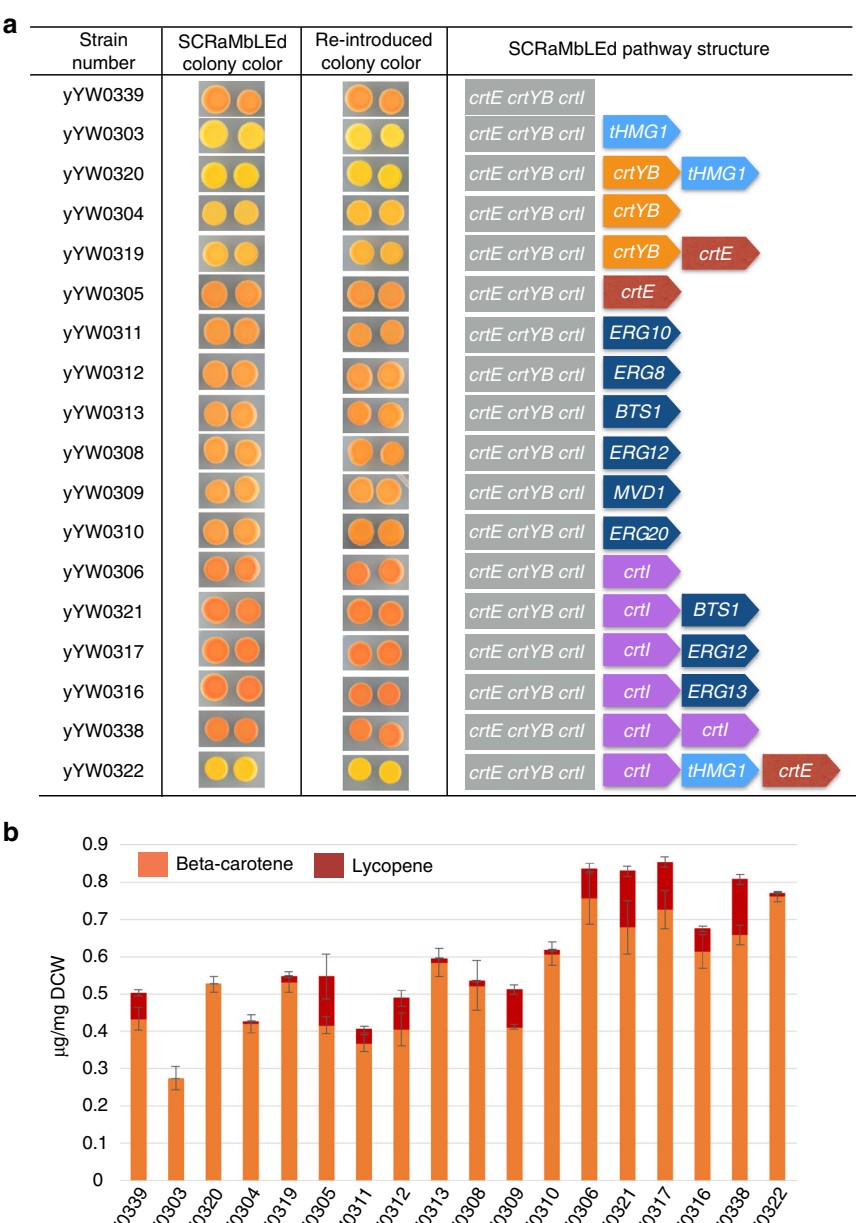

**Fig. 4** Bottom-up in vitro SCRaMbLE for β-carotene pathway optimization. **a** Phenotype–genotype correlation of the β-carotene pathway after bottom-up in vitro SCRaMbLE (strategy 1). A total of 17 SCRaMbLEd yeast strains were isolated for testing colony color and determination of SCRaMbLEd construct sequences. The pictures of yeast color for both SCRaMbLEd strains and strains with re-introduced SCRaMbLEd plasmid were taken after 3 days incubation at 30 °C on SC–Ura medium. The SCRaMbLEd pathway structure was initially analyzed by restriction enzyme digestion to check the number of insertions and PCR analysis to evaluate which genes were inserted. Primer walking sequencing was applied to verify the sequence of all recovered plasmids. yYW0339 with only a URA3 gene inserted was used as a control strain. **b** HPLC measurement of carotenoid production for SCRaMbLEd yeast strains. Quantification was performed in biological triplicate for each strain as shown. Error bars represent standard deviation from three replicates

SCRaMbLEd endogenous TU pool (Fig. 3c), suggesting that varying the copy number of genes in the endogenous mevalonate pathway has no major impact on β-carotene pathway productivity, based on visual inspection.

To test the insertion efficiency for the first version of bottom-up SCRaMbLE, we performed the in vitro Cre reaction with different ratios of acceptor vector and donor fragments. Reaction products were evaluated after transformation into yeast. Most of the SCRaMbLEd yeast strains (70–85%) carried a single insertion. Increasing the donor fragment: acceptor vector ratio by ten-fold nearly doubled the number of times we observed two or more insertion events, from 12% to ~20% (Fig. 3d). In fact, we were able to assemble the entire four-gene beta carotene pathway in a single bottom-up in vitro SCRaMbLE experiment using this strategy (Supplementary Fig. 11).

Single colonies of diverse colors and intensity were randomly streaked out to interrogate the inheritance of color formation. After yeast colony PCR analysis with TU specific primers in 100 randomly picked colonies, 17 strains showed diverse SCRaMbLEd structures (Supplementary Fig. 12). To verify colony color was dependent on SCRaMbLEd plasmids, the yeast plasmids were recovered into *E. coli* and then re-transformed into the parental yeast strain yYW0246. All re-transformed strains developed the identical colony color compared with original SCRaMbLEd isolates (Fig. 4a). Yeast colonies with diverse pathway structure were analyzed for production of β-carotene and lycopene, determined by HPLC (Fig. 4b).

Phenotype–genotype correlation of β-carotene pathway indicated that an additional copy of *crtI* gene led to a deep orange colony color and increased production of β-carotene (compare strains yYW0306 with yYW0339, Fig. 4). On the other hand, strain yYW0338 with two additional copies of the *crtI* gene did not increase production of β-carotene beyond that observed in yYW0306, indicating one additional copy of *crtI* is sufficient to optimize the β-carotene pathway. An additional copy of *tHMG1* can make the colony color bright yellow (yYW0303, yYW0320, yYW0322) and produces an HPLC profile very similar to that of purified β-carotene (Supplementary Fig. 13). Interestingly, an additional copy of *crtYB* can make the colony color deep yellow (yYW0304, yYW0319) but in this case two unknown peaks appeared after the peak of β-carotene, which are presumably caused by production of other carotenoids[19] (Supplementary Fig. 13).

In the first version of bottom-up SCRaMbLE, because each donor fragment encodes a *URA3* gene, it can lead to instability in constructs with multiple TUs incorporated. We performed an experiment to test the stability of in vitro SCRaMbLEd constructs carrying two and three TUs, yeast strains yYW0320 and yYW0322, respectively. After continuous passage for 100 generations, we identified instances of recombination, 1/159 for yYW0322 and 5/120 for yYW0320 (Supplementary Fig. 14). This is obviously undesirable for any kind of production application. This problem is circumvented with the second version of bottom-up in vitro SCRaMbLE (Fig. 3a, right panel), which yields recombined products lacking direct repeats and additionally enables counter-selection to remove background unmodified acceptor vector.

## Discussion

With the rapid development of DNA synthesis and assembly technology, there are a growing number of researchers using synthetic DNA for de novo design and construction of heterologous pathways and synthetic genomes. Inserting 34 bp loxPsym sites in the 3′untranslated region (UTR) of nonessential genes in synthetic yeast chromosomes or at the boundary of transcription units has shown no detectable impact on the expression of neighboring genes[3,25,26]. The addition of loxPsym sites provides genetic flexibility and enables chromosome or pathway rearrangements mediated by Cre recombinase[13,27].

In this study, we demonstrated top-down in vitro SCRaMbLE for construction of pathway structural variation library as applied to pathway flux optimization. Compared with other in vitro recombination methods, which mainly focus on single recombination events[28–30], the top-down in vitro SCRaMbLE system achieves combinatorial rearrangements precisely between carefully placed loxPsym sites to yield complex new genetic architecture in a loxPsym-enabled pathway or chromosome. Unlike random mutagenesis, the SCRaMbLE system uses functional modularity as the basic building block of variation, via copy number variation, as well as changes to TU order and orientation. The diversity of the SCRaMbLEd DNA pool partly relies on the number of loxPsym sites in the initial construct. The more building blocks that are involved, the more diverse the resulting SCRaMbLEd pool. The top-down in vitro SCRaMbLE system is a convenient way to generate combinatorial diversity of DNA constructs with no need for selectable markers.

The SCRaMbLE system promotes deletion, inversion, and duplication events, making it an interesting tool for studying evolution, in particular duplication events could readily lead to a gain of function[31]. In our 100 kb synIXR-BAC in vitro SCRaMbLE experiment, >70% of transformed *E. coli* cells showed new combinatorial structures (Supplementary Fig. 5d). However, because the Cre recombinase reaction goes to equilibrium, the frequency of cells carrying SCRaMbLEd sequences may be lower when there are fewer loxPsym sites in the initial constructs. Using a yeast centromere plasmid encoding the β-carotene pathway genes as an example, we demonstrate the in vitro SCRaMbLE system can be used to optimize biosynthetic pathway flux via rearrangement of pathway TUs. The production of β-carotene in yeast can be increased by duplication and inversion of *crtI* gene in the constructed pathway[18] (Fig. 2). The top-down in vitro SCRaMbLE method provides a high throughput way to reconstruct pathway structures. This is particularly useful to study genetic networks and gene interactions.

To circumvent the need to assemble a multi-TU pathway encoding loxPsym sites for top-down in vitro SCRaMbLE, we developed bottom-up in vitro SCRaMbLE. Using the β-carotene pathway as an example, we observed that the recombined DNA pool yielded diverse carotenoid production in yeast. The production of β-carotene was increased and fewer carotenoid intermediates were observed with additional copies of the *crtI* and *tHMG1* genes. For strategy 1 (*URA3* marker in the donor fragments), we observed insertion of two or three donor fragments into the acceptor vector with a > 10% frequency. This ratio was increased (up to ~20%) by increasing the mole ratio of donor fragments to acceptor vector.

Together, the top-down and bottom-up in vitro SCRaMbLE systems provide an efficient strategy to generate rearranged and optimized genetic structures. We have demonstrated that in vitro SCRaMbLE has several advantages over the in vivo method. 1) In vitro SCRaMbLE is highly controllable as compared to the in vivo reaction; while the in vitro reaction can be stopped by heat inactivation, leaky Cre activity in vivo is a known problem and can lead to pathway and genome instability[13,14]. 2) One can isolate sub-libraries with varied numbers of deleted building blocks by gel purification of digested SCRaMbLEd pools (Supplementary Fig. 2). Here the efficiency of in vitro SCRaMbLE could be further optimized by identifying loss of a restriction enzyme cut site in the SCRaMbLEd construct (Supplementary Fig. 15). 3) In vitro SCRaMbLE reactions reach equilibrium in 10 min and are stable for 16 h (Supplementary Fig. 16), whereas the

in vivo reaction depends on ongoing expression of Cre recombinase. Deletion events accumulate with longer SCRaMbLE time, which can lead to reduced library complexity. 4) The phenotype–genotype analysis of in vitro SCRaMbLE is easier and more straightforward than in vivo because of less noise from the genome of host strains. 5) The in vitro SCRaMbLEd pool can be transformed into different host strains, further expanding the applicability of this method.

## Methods

**Strains and plasmids**. These are described in Supplementary Tables 1 and 2.

**Construction of loxPsym site plasmids**. The pathway encoded in pLM495 was initially assembled using VEGAS (versatile genetic assembly system)[19], and loxPsym sites were subsequently introduced between each pathway gene through PCR reactions using primers that introduced loxPsym sites and terminal, inward pointing *Bsa*I sites. pLM495 was then assembled by Golden Gate. The ~100 kb synIXR-BAC was previously described[14]. pYW0261 was assembled from 500 bp sectional sequences randomly chosen from β-carotene pathway genes (*BTS1*, *crtE*, *crtI*, *crtYB*, *ERG8*, *ERG10*, *ERG12*, *ERG13*, and *ERG20*), respectively, and interspersed with loxPsym sites.

**Construction of acceptor vector and donor fragments**. Acceptor vector pYW0113 is a yeast centromere plasmid with a *HIS3* gene as auxotrophic marker and a red fluorescent protein (RFP) gene flanked by two loxPsym sites. Donor universal vector pYW0120 was assembled using stepwise PCR to introduce "*Not*I-loxPsym-*Bsa*I-RFP-*Bsa*I-URA-loxPsym-*Not*I" structure to a high copy *E. coli* plasmid backbone. The donor transcription units were amplified with primers that introduced terminal *Bsa*I restriction sites, which were subsequently assembled into the universal vector pYW0120 by Golden Gate assembly. All donor fragments "TU + URA" were obtained by *Not*I digestion followed by gel purification.

**synIXR-BAC isolation**. synIXR-BAC DNA was prepared using standard alkaline lysis and ethanol techniques[32].

**In vitro SCRaMbLE**. The Cre recombinase reaction was set up as per the manufacturer's instructions (NEB, M0298L) and incubated at 37 °C for 1 h. The Cre enzyme was heat inactivated for 10 min at 70 °C. For top-down in vitro SCRaMbLE, 100 ng of DNA was added in a total reaction volume of 10 μl with 1 μl of Cre recombinase. For bottom-up in vitro SCRaMbLE, 200 ng acceptor vector was mixed with the donor fragments pool (1000 ng in total) in a reaction volume of 50 μl with 1 μl of high concentration Cre recombinase (NEB, M0298M). Both SCRaMbLEd pools were transformed to hosts for genotype and phenotype testing. For bottom-up in vitro SCRaMbLE, SC–Ura–His medium or SC–His + 5-FOA medium are used to select for recombined constructs, depending whether the first or second version is used.

**Yeast plasmid recovery**. SCRaMbLEd plasmids were recovered from yeast using the following method. A volume of 1.5 ml overnight cultured yeast cells were collected and resuspended in 250 μl of P1 (Qiagen) with 10 mg per ml RNase and 200 μl glass beads followed by shaking for 10 mins to mechanically break open the cells. Then plasmids were isolated using with the standard alkaline lysis and a Qiagen miniprep spin column to isolate the DNA. The plasmids were eluted with 30 μl of elution buffer. 15 μl of the elution was transformed to 100 μl of *E. coli* competent cells.

**Plasmid structure determination**. Methods to analyze recovered plasmids included restriction digestion analysis, PCR analysis with gene specific primers, Sanger sequencing, and PacBio sequencing. The top-down SCRaMbLEd plasmid pYW0108 with duplicated genes was initially analyzed using restriction digestion and then sent for PacBio sequencing. Other top-down SCRaMbLEd plasmids were initially analyzed using restriction digestion and subsequently analyzed using a primer walking sequencing method. All bottom-up SCRaMbLEd plasmids were initially analyzed by restriction digestion to check the number of insertions and PCR analysis to identify the inserted gene. Primer walking sequencing was applied to verify all the recovered bottom-up SCRaMbLEd plasmids.

**ARTP of yeast strains**. The yeast strain yYW0257 with $OD_{600}$ value at 2 was selected to undergo ARTP. The RF power input was set to 120 W and the temperature of the plasma jet was set to 25–35 °C. Ten microlitres of the cell culture was dipped onto the stainless steel minidisc and then exposed to ARTP jet for 0 s (control), 10 s, 20 s, 30 s, respectively. Then the treated yeast cells were diluted in the selective medium. This was done on ARTP-II device from Wuxi Research Institute of Applied Technologies (Wuxi, China).

**PacBio sequencing of SCRaMbLEd library**. The analyzed library was derived from a DNA pool of in vitro SCRaMbLEd pLM495 by linearizing with *Not*I and *Sal*I. The library was sequenced on an RSII sequencer from Pacific Biosystems (Menlo Park, CA, USA). The alignment was performed with software BLAST.

**HPLC measurement of carotenoid production**. SCRaMbLEd yeast strains and control yeast strains were cultured in 5 ml of SC–Ura liquid medium at 250 r.p.m., 30 °C in a shaking incubator. The saturated cultures were diluted to an initial $OD_{600}$ of 0.1 in 50 ml of SC–Ura liquid medium and grown for 48 h with the same condition. An aliquot of the culture was centrifuged for 5 min at 4000 g. Cells were resuspended in 1 ml of 3 M HCl. The resuspended cells were heated in a boiled water bath for 3 min, and then cooled in an ice-bath for 3 min, repeating twice. Cell pellets were then washed twice with double-distilled water and harvested by centrifugation. After removal of the supernatant, the cells were resuspended in 1 ml acetone and vortexed for 10 min. The acetone extracts were centrifuged and filtered with a 0.22 μm filter for subsequent analysis. A portion of each sample was harvested and dried at 70 °C for measurement of the dry cell weight. The analysis of carotenoids was performed by HPLC (Waters 2695) equipped with SUPELCO C18 column (33 cm × 4.6 mm) and UV detection at 450 nm and 470 nm. The mobile phase consisted of acetonitrile-methanol-dichloromethane (18:90:2 v/v) with a flow rate of 0.3 ml per min at 25 °C. The content of the carotenoids was expressed as μg per mg dry cell weight . Each samples were performed on technical triplicates.

**Data availability**. The PacBio sequencing data has been deposited at BIG Data Center (http://bigd.big.ac.cn/) with accession code 'CRA000752'.

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

## Acknowledgements

This work was funded by the National Natural Science Foundation of China (21750001, 21621004, and 21676192), the Ministry of Science and Technology of China ("973" Program, 2014CB745100), and the International S&T Cooperation Program of China (2015DFA00960), and US research was funded by NSF grants MCB-1026068 and MCB-1158201. We thank Andrew Martin, Yu Zhao, Mingzhe Han, and Sijie Zhou for their technical support.

## Author contributions

Y.W., Y.Y., and J.D.B. conceived the study and designed experiments; Y.W., R.Z., L.A.M., L.M., R.L., M.Z., B.J., H.X., Y.L., Z.Y., and Y.M. performed experiments; Y.W., R.Z., L.A. M., X.L., H.L., D.L., W.X., X.Z., and B.L. analyzed data; and Y.W., L.A.M., Y.Y., and J.D. B. wrote the paper.

## Additional information

**Competing interests:** J.D.B. is a founder and director of Neochromosome, Inc., and CDI Labs, Inc. J.D.B. serves as a scientific advisor to Recombinetics, Inc., Modern Meadow, Inc., and Sample6, Inc. These arrangements are reviewed and managed by the committee on conflict of interest at NYU Langone Health. The remaining authors declare no competing interests.

