## [Peer Review File(PDF 195 kb) · Nature Communications]

REVIEWERS' COMMENTS:

Reviewer #1 (Remarks to the Author):

The revised work by Wu et al. carefully addresses the previous reviewer concerns in a more comprehensive and impactful paper. I believe the method has the potential to be readily picked up by other labs, especially since *in vitro* SCRaMBLE has several advantages over the *in vivo* method. While I would love to have seen them apply this system to a larger metabolic pathway, or genetic circuit, I am supportive of the work.

Reviewer #2 (Remarks to the Author):

In this revised manuscript, the authors have addressed most of the concerns raised in the previous review. However, a minor revision is still required. The following points should be addressed during revision.

1. It is unclear to me whether a clearer description of the term "top-down and "bottom-up" was added in the revised manuscript.
2. "E.coli" should be changed to "E. coli" throughout the manuscript.
3. In section Results: in paragraph 2, "5kb" should be changed to "5 kb".

Reviewer #3 (Remarks to the Author):

The authors have done an excellent job in addressing the reviewers comments. The authors have done additional experiments to address the reviewers concerns, have reduced the explanation in the introduction, and moved a figure to supplementary materials.

One minor point. The authors refer to *loxP* sites in the introduction but don't really define them. It would be good if they defined them at the start. Additionally, it would be helpful if they highlighted the differences between the *loxP* and *loxP*_{sym} sites in Figure 1.

This reviewer has not further concerns.

REVIEWERS' COMMENTS:

Reviewer #1 (Remarks to the Author):

The revised work by Wu et al. carefully addresses the previous reviewer concerns in a more comprehensive and impactful paper. I believe the method has the potential to be readily picked up by other labs, especially since in vitro SCRaMbLE has several advantages over the in vivo method. While I would love to have seen them apply this system to a larger metabolic pathway, or genetic circuit, I am supportive of the work.

Response: Thanks for your comments. We are planning to apply in vitro SCRaMbLE on a larger metabolic pathway in a future study.

Reviewer #2 (Remarks to the Author):

In this revised manuscript, the authors have addressed most of the concerns raised in the previous review. However, a minor revision is still required. The following points should be addressed during revision.

1. It is unclear to me whether a clearer description of the term “top-down and “bottom-up” was added in the revised manuscript.

Response: It was added in paragraph 4 of main text.

2. "E.coli" should be changed to "E. coli" throughout the manuscript.

Response: Fixed.

3. In section Results: in paragraph 2, "5kb" should be changed to "5 kb".

Response: Fixed.

Reviewer #3 (Remarks to the Author):

The authors have done an excellent job in addressing the reviewers comments. The authors have done additional experiments to address the reviewers concerns, have reduced the explanation in the introduction, and moved a figure to supplementary materials.

One minor point. The authors refer to loxPsym sites in the introduction but don't really define them. It would be good if they defined them at the start. Additionally, it would be helpful if they highlighted the differences between the loxP and loxPsym sites in Figure 1.

This reviewer has not further concerns.

Response: The definition of loxPsym site is in the beginning of paragraph 3 in main text, and the figure 1b highlighted the differences between the loxP and loxPsym sites.